# Mental Health Challenges and the Associated Factors in Women Living with HIV Who Have Children Living with HIV in Indonesia: A Qualitative Study

**DOI:** 10.3390/ijerph19116879

**Published:** 2022-06-04

**Authors:** Nelsensius Klau Fauk, Maria Silvia Merry, Lillian Mwanri, Karen Hawke, Paul Russell Ward

**Affiliations:** 1Research Centre for Public Health Policy, Torrens University Australia, Adelaide, SA 5000, Australia; nelsensius.fauk@torrens.edu.au (N.K.F.); lillian.mwanri@torrens.edu.au (L.M.); 2Institute of Resource Governance and Social Change, Kupang 85227, Indonesia; 3Medicine Faculty, Duta Wacana Christian University, Yogyakarta 55224, Indonesia; silvia.tropmed@yahoo.com; 4Infectious Disease—Aboriginal Health, South Australian Health and Medical Research Institute, Adelaide, SA 5000, Australia; karen.hawke@sahmri.com

**Keywords:** mental health challenges, supporting factors, mothers living with HIV, children living with HIV, Indonesia

## Abstract

Women living with HIV (WLHIV) are vulnerable to various mental health challenges. However, there is a paucity of studies globally and in the Indonesian context that have specifically explored mental health challenges among mothers living with HIV who also have children living with HIV (CLHIV). This qualitative study explored mental health challenges and the associated factors in mothers living with HIV who have CLHIV in Yogyakarta, Indonesia. In-depth interviews were employed to collect data from the participants (N = 23) who were recruited using the snowball sampling technique. A qualitative data analysis framework was used to guide the analysis of the findings. The findings showed that the mothers experienced a range of mental health issues due to their own, and their child’s diagnoses of stress, depression, anxiety, fear, sadness, and guilt. Lack of knowledge about HIV, fear of death, shame, not knowing whom to talk with and what to do after their own HIV diagnosis, and the HIV diagnosis of their children were factors that challenged their mental health. Difficulties in dealing with daily life or social activities of their CLHIV, dilemmas in addressing questions and complaints of their CLHIV about HIV treatment, and concerns about the health condition of their CLHIV and how their children cope with any potential negative social impacts also impacted the mothers’ mental health. Social factors such as unsympathetic expressions from friends towards them and their CLHIV and negatively worded religious-related advice from parents and relatives also contributed to their poor mental health. Our findings indicate the need for intervention programs that support mothers living with HIV and their CLHIV. Future large-scale studies involving mothers living with HIV who have CLHIV in Indonesia and other settings globally are needed to obtain a comprehensive understanding of mental health challenges and the associated factors they face.

## 1. Introduction

The UNAIDS reports an estimated 37.7 million people living with HIV worldwide in 2020, of which over half (53%) are females aged 15 and over [1]. The same report also shows a high prevalence of HIV infection for females in sub-Saharan Africa, where girls and women living with HIV (WLHIV) account for 63% of total infections in the region [1,2]. Females in the same age category also represent 30% of the 5.8 million people living with HIV across Asia and the Pacific region [1,2]. In the context of Indonesia, women aged 15 years and over represent 38% of the 427,201 people living with HIV in 2021 [2,3]. Mothers and/or housewives are not only a high-risk group for HIV in Indonesia, but also do suffer the most from the impacts of HIV [3]. The Indonesian national AIDS report shows that for the last 10 years, more than 1000 HIV-positive mothers and/or housewives progressed to AIDS annually [3]. They are the second highest group of people living with AIDS in the country, comprising 14.3% of the total AIDS cases, following unskilled workers, who account for 16.2% [3].

Existing evidence has suggested that HIV in women in general leads to a range of mental health challenges, including stress, anxiety, depression, sadness, and embarrassment [4,5,6,7,8,9]. The stressors for these mental health issues in WLHIV vary and include advanced stage of HIV infection, weakened physical condition, and the fear of a breach of the confidentiality of their HIV status, which may cause further negative impacts on themselves and their family [8,9,10,11,12,13]. Other stressors include fear of transmitting HIV to their unborn babies, the burden of being a caretaker for their children, concerns about their future and the future of their children, and a lack of resources needed to support their children and family [5,13,14,15,16,17,18]. Similarly, the lack of social support, the experience of social rejection and social isolation, avoidance by family members, internalised or perceived stigma [5,9,13,15,19,20], and poor economic conditions [7,10,16] are also reported determinants for depression, anxiety, fear, and worry among WLHIV. Additional factors such as low level of education attainment, cessation of a relationship with a partner or divorce, a partner’s death, family misfortune, family food insecurity, and treatment failure [7,10,12,19,20], are all predictors of negative impacts to mental health challenges.

Despite this, there is a lack of research globally and within Indonesia about HIV-related mental health challenges and the associated factors in mothers living with HIV who have CLHIV. Whilst a few studies on WLHIV in the context of Indonesia may have overlaps with mothers living with HIV as some of the WLHIV may also be mothers [21,22,23], none of them has focused on mental health challenges experienced by these women. This study aims to fill this gap in knowledge by interviewing mothers living with HIV who also have CLHIV in Yogyakarta, Indonesia. Yogyakarta was chosen as the study setting due to feasibility, familiarity, and the potential of undertaking the current study successfully. Exploring these factors is imperative to form an understanding of the lived experience of mothers living with HIV and their children, to better inform the development of policies and intervention programs that address the complex needs of both the mothers and the CLHIV.

## 2. Methods and Analysis

This study was carried out in Yogyakarta municipality, which is one of five districts in the Special Region of Yogyakarta province, Indonesia. It covers an area of 32.50 Km^2^ and has a total population of 402,679 people [24]. In terms of HIV cases, the 2021 report suggests that it has a total of 1422 cases, which are higher than those of four other districts in the province [25]. Women account for 29.9% of the total number of HIV cases in this area, and of these women, approximately one quarter, have become AIDS sufferers [25,26]. There are several HIV care services provided for people living with HIV in this area such as HIV information sessions; antiretroviral medicines; voluntary counselling and HIV testing; and several medical tests, including CD4, viral load, liver and kidney function tests [27]. These services are provided 4 hospitals and 10 community health centres in this area [27].

A qualitative methodology was used to explore mental health impacts of HIV on mothers living with HIV who have CLHIV. The study focused on understanding mental health challenges and the associated factors they faced following their own and their children’s HIV diagnosis. We recruited the participants using the snowball sampling technique. Initially, we solicited the help from the head of a HIV clinic and a buddy of people living with HIV (also known as pendamping orang dengan HIV/AIDS), who is also living with HIV, in the study setting. Both agreed to help distribute the study information sheets containing contact details of the researchers to potential participants. The information sheets were then posted on the information board at the clinic by the clinic receptionist and on the WhatsApp group of people living with HIV by the buddies of people living with HIV. Potential participants who contacted the researchers and confirmed their participation were recruited for an interview. After the interviews, initial participants were also asked for help to distribute the study information sheets to their eligible friends who might be willing to participate in this study. None of them who texted or called to confirm their participation withdrew from the study. They were recruited based on several inclusion criteria: one had to be 18 years old or above, and a mother living with HIV who has a child or children living with HIV. Finally, we recruited and interviewed 23 participants. The participants’ age ranged from 24 to 43 years old. The majority graduated from senior and junior high school (see Table 1) and were engaged in different kinds of professions, while five were housewives and three were unemployed. They had all been living with HIV for different periods of time. All participants and their CLHIV were on antiretroviral therapy (ART) when the interviews were conducted. The ART for people living with HIV, including these mothers and their children, in Indonesia was fully subsidised by the national government of Indonesia [28]. In addition, these mothers and their children held the National Health Insurance, commonly known as the Social Insurance Administration Organisation (Badan Penyelenggara Jaminan Sosial or BPJS) [29]. This was either the fully government-subsidised type, commonly known as Indonesian Health Card (Kartu Indonesia Sehat or KIS), or the independent (mandiri) type, which required them to pay a certain amount of monthly insurance fee (±USD 2–5), which depended on the level they chose [29]. This BPJS insurance enabled them to have free access (including free administration fee at healthcare facilities) to both general health care and HIV-related care services [29]. Each mother had only one child living with HIV and nine mothers had an additional child or children who are not living with HIV. The details of their CLHIV are presented in Table 1.

In-depth interviews were carried out to collect the data from the participants in Yogyakarta, Indonesia. Interviews were conducted face-to-face and online using WhatsApp video calls. These modes of interview and interview times were decided based on the preference of the participants. Interviews explored participants’ views and experiences of mental health challenges they faced following their HIV diagnosis; their views and experiences of how their children’s HIV diagnosis influenced their mental health condition; their experience of dealing with or managing social activities, social life, health condition, and treatment of their CLHIV; and how these factors influenced their mental health condition; the interviews also explored their views about the influence of their social relations with families, relatives, and friends on their mental health condition. The duration of the interviews was approximately 40–60 min, with only an interviewer and each participant present during the interviews. Interviews were conducted in Indonesian, the primary language of the interviewers (NKF, MSM) and the participants, and were recorded. Sometimes notes were also taken during the interviews, if felt necessary. Interviews ceased when we felt that data had been rich enough to address the research questions and aim, and that data saturation had been reached and no new information was obtained from the responses of the last few participants. At the end of the interviews, we offered each participant an opportunity to read and correct the information they had provided after the transcription of the recordings, but none took the opportunity. The participants were assured that information they provided would remain confidential and anonymous, and a pseudonym was assigned to each interview transcript. They were also advised prior to the interviews that they had the rights to withdraw from the study for any reason, at any time. Each participant signed an informed consent to indicate their willingness to participate and returned it in person or via WhatsApp prior to or on the interview day.

The comprehensive data coding and analysis were performed manually following the verbatim transcription of the interview recordings. Transcription occurred alongside the data collection process, which enabled the direct integration of notes taken during the interviews into each transcript. Coding and analysis were conducted in Indonesian and quotes for publication purposes were translated into English by NKF who is fluent in both Indonesia and English. To maintain the accuracy of translation, checking and rechecking the original transcripts against the translated interpretations were repeatedly conducted during data analysis process [30]. Informed by Ritchie and Spencer’s qualitative data analysis framework, five steps were undertaken throughout data analysis process [31,32]. These included: (1) familiarisation with the data, which was performed by reading each individual transcript repeatedly. While reading the transcripts, the participants’ narratives were broken down into chunks of information, which were then labelled and commented to search for patterns and connections. (2) Identification of a thematic framework, which was conducted by identifying and listing key issues, concepts, and ideas that emerged from the transcripts, which were then used to form a thematic framework. The identification of the thematic framework involved changing and refining the issues, concepts, and ideas throughout data analysis process. (3) Indexing the entire data through which data extracts in each individual transcript were indexed or coded (open coding) resulting in a long list of codes. This was followed by the identification of similar or redundant codes, which were then collated (close coding) to reduce the list of codes to a manageable number. Codes that formed a logical pattern to explain a theme were grouped together under the theme or sub-theme. (4) Charting the data through which all themes and their codes that had been created in the previous steps were reorganised in a summary of chart for the purpose of comparison within and across the interviews, (5) Lastly, the entire data were mapped and interpreted. This data analysis was non-linear and an iterative process that involved changing and refining codes and themes before we agreed upon the final themes presented in this paper. The steps undertaken in this analysis helped us manage the data in a structured and coherent manner and enhanced the rigour, transparency, and validity of the analytical process [31,32].

Ethics approval for this study was obtained from Health Research Ethics Committee, Duta Wacana Christian University (No. 1005/C. 16/FK/2019).

## 3. Results

### 3.1. The Experience of Mental Health Challenges

Mental health challenges were experienced by all the study participants after their HIV diagnosis. They described that it led to *“extreme stress and depression for a few years”* (Ayu) and *“feeling sad, afraid and anxious every day at the beginning of the diagnosis”* (Indah). Lack of knowledge about HIV, fear of death, shame, and not knowing anybody to talk with and what to do following the diagnosis were some of the contributing factors for the feelings of stress, depression, anxiety, fear, and sadness they experienced. The following narrative of a participant who had been diagnosed with HIV for nearly 5 years illustrates how such factors influenced her feelings after her HIV diagnosis:


*“After being diagnosed with HIV what I could do was crying. I was very stressed, sad, scared, all mixed up. I didn’t know anything at all about HIV, haven’t heard of it before. I was very scared because I thought I was going to die soon. I didn’t know whom to ask for help, whom to talk with about this problem. I didn’t know what to do. I didn’t dare to seek help from other people because I was ashamed if people know my HIV status. I remember I was so depressed. So, for at least two years I struggled with those feelings myself”*
(Julia)

The HIV diagnosis of their child was another factor that impacted their mental health significantly. They acknowledged that it had increased the burden on them and put them into a situation where they often felt guilty *“Since my second daughter tested positive with HIV, I feel very much guilty. When she is sleeping, I often look at her and cry, and say ‘sorry mom has made you suffer from this [HIV]* (Anti)”. They also experienced prolonged worry *“I am worried all the time about my daughter since she tested positive last year. I have been thinking too much about her”* (Dewi). The stories of the participants also indicated that once their child was diagnosed, there was a definite shift in their focus, with much of their attention and thoughts going to their CLHIV. The following narrative of a single mother who has a daughter living with HIV depicts such experiences:


*“After I received the [HIV] test result of my daughter, I told myself that I must be strong and focus on the life of my daughter. But you know what, I feel the burden is even heavier. These feelings: afraid, sad, worried, stressed, guilty, don’t go away. I feel these up to now, but not because I am caring about myself. It is all about my daughter. I think a lot about her and put all my attention and focus on her. I fear any negative impacts she may face in the future due to this [HIV] and it makes me sad and stressed out. It is my mistake that I transmitted HIV to her, she doesn’t deserve this. The more I think about her situation and future the more worried and stressed I get. I have been trying to live a normal life without thinking too much but these thoughts and feelings never go away”*
(Jeane)

### 3.2. Daily Life Challenges Associated with Their Children’s Condition of Living with HIV

Dealing with daily life or social activities of their CLHIV was a significant challenge for the mothers. They described that their children’s daily activities, such as going to school, mingling with friends, and engaging extracurricular activities with other students including school excursions, picnics, and sport, increased their fear and worry about the possibility of their children’s HIV status being disclosed or discovered. Yeni, a mother of a 12-year-old son living with HIV, stated “*I am worried every day because my son often mingles and plays football with his friends. On the one hand, I do not feel right to forbid him to play with his friends, but on the other hand I am worried if his friends find out about his status*”. Similarly, Enda, who has a 10-year-old daughter living with HIV, described “*I do not feel calm and am always worried every day I walk my daughter to school or when she does her homework at her friends’ house. I am afraid if her friends or their parents are suspicious and find out that she has HIV*”. Such over thinking and fear experienced by the mothers seemed to be influenced by their projection of negative impacts that could happen to their children if they disclosed to their friends and/or other people their HIV status. The following quote illustrates such assertions:


*“My daughter is now in junior high school. She sometimes goes out with her friends. They do their homework together. Sometimes, she goes to her friends’ place, or her friends come to our house. I am happy that she has friends but deep inside I have always been worried and afraid if she accidently tells her friends about her condition. I am scared thinking of any possible negative impacts that could happen to her if other people know about her HIV status. Her friends may avoid or reject her and this could make her down, stressed and unwilling to go to school. All the activities she engages in always bring me these kinds of thoughts and fears. Sometimes, I want to limit her activities but I can’t do that because I see her happy doing what she does. It hasn’t been easy for me to deal with her daily life and activities….”*
(Betty)

Most mothers in the study acknowledged that their CLHIV were not aware or had not been informed about their HIV status because of their young age. This complicated the mothers’ experience of living with HIV and they found it very challenging managing their children’s health condition, especially when their children disliked medications and questioned why they needed to take them regularly. For example, the children would ask: “*Why should I take this [ART] medicine every day*?” (Silvia), “*Can I skip taking the medicine today*?” (Astri), “*I don’t want to take the medicine*” (Retno). This questioning, complaining, and rejection by their children complicated the prevailing situations, compounding the impact of HIV on mental health of these women:


*“I just don’t know how to tell my son [about his HIV status] because he is still little, eight years old. He can’t understand the complex issue around HIV. He may go and tell his friends that he has HIV without knowing the negative consequences he may face. No one can guarantee that he will keep it secret. It has been very stressful for me because he keeps on asking why he needs to take the medicine every day. He complains a lot and sometimes cries and rejects to take the medicines. I feel guilty for what he is going through and will be facing in his entire life”*
(Fatima)

Concerns about what action they need to take if their children get sick and how to prepare their children to deal with their health condition and the surrounding social environment in the future were also factors that made them feel stressed and scared. Yanri, a mother who has a 9-year-old son living with HIV, described “*We [the woman and her HIV-positive husband] have always been trying to prepare ourselves and think of what to do if our son falls ill, and this is stressful considering my situation of not having a permanent job or purely relying on my husband*”. She continued “*I am also concerned with how he will deal with his condition and people around him when he knows that he is HIV-positive. I want to prepare him for these and I feel scared*”. Such concerns seemed to also be influenced by their unstable economic or financial conditions and negative treatments towards their children or other children living with HIV by uninfected people within families, communities, and healthcare settings:


*“I am concerned with my daughter’s health condition. I see that she is not very strong and can get sick anytime. What makes me stressed out is that I don’t have a job, no income at all. I always think of what I am going to do if she gets sick or is admitted to hospital. So far, we [the woman and her daughter] rely on financial support from my parents. The situation I am in now makes me worried. To be honest I am scared deep inside. I have just submitted an application for a job a few days ago, hopefully I get accepted”*
(Kana, a single mother, her husband died from AIDS)


*“We [the woman and her HIV-positive son] experienced negative treatment once at the hospital. We were not served; my son was not allowed to use the toilet at the hospital. It was three years ago, he was four years old so he didn’t really understand. That is why I am concerned with how to prepare him to face such kinds of negative treatments. There are many kids [living with HIV] who receive negative treatments from their friends at school or within communities. I don’t want him to feel broken if these happen to him, that is why I am worried”*
(Ratih)

### 3.3. Social Factors: Unsympathetic Expressions and Comments from Others

Social factors such as unsympathetic expressions and comments from friends towards participants’ children living with HIV negatively impacted their mental health condition. A few participants, who acknowledged having told their close friends about their own and their children’s HIV status, described that some of their friends seemed to be trying to show caring attitudes or attentions towards CLHIV, but their expressions and comments sometimes made them feel guilty, attacked, and stressed. The quote presented below reflects such kinds of expressions or comments they received from their friends:


*“There are some close friends of mine whom I have told about my own and my son’s situation because they have been nice to us. But sometimes their words which seem to reflect their sympathy for my son but actually pierce my heart. For example, one of them said to my son ‘You are a strong child, keep your spirit up, take medicine. It’s not your fault, you should be fine’. In other words, it’s your mother’s fault. I feel accused and guilty. I sometimes feel stressed out when I remember other people’s words in this tone. After I heard what they said I try to keep distance from them, we are not that close anymore”*
(Shinta)

Similarly, unsympathetic comments from friends towards the participants themselves were acknowledged as having an influence of their mental health condition. The following story of a woman, who acknowledged contracting HIV from her ex-boyfriend prior to her marriage with her current husband, reflects unsympathetic comments of her friends that made her feel shocked and depressed:


*“Some of my friends, who know that we [the woman and her daughter] are HIV-positive, have negative comments about me. Some said ‘This is a consequence of her own actions. She had sex with several men before marriage, that is why she got HIV. It’s a pity that her little daughter must also bear the consequences’. When a very close friend of mine told me this [what her other friends said about her], I was shocked and depressed because I never thought that they had such a negative view about me”*
(Ima)

### 3.4. Religious Advice from Family Members

Religious advice received from parents and relatives were also reported by some participants to bring back their memories of some difficult past experiences they had gone through. Some described that their parents’ religious advice reminded them of the mistakes they had committed in the past, and these were acknowledged as not helping but making them feel guilty, angry, and stressed. The following story from an unmarried woman and a mother of two daughters, of whom one is living with HIV, reflects such experiences:


*“My mother and father always give me advices. I remember my father said ‘You have made mistakes and what you are now going through is a warning from God for you. Now you should try to get closer to God…’. I understand what they mean is that I have been guilty of having children before marriage or not having a husband. This is wrong because it is not in accordance with our [Islamic] religious teachings. As a result of my mistakes, my daughter and I got HIV. This seems like a good advice but it makes me feel guilty and stressed out. Advice from parents sometimes makes me get mad at myself and what I have done”*
(Rini)

Similar comments were raised by Lina, a 28-year-old mother who acknowledged to engage in injecting drug use and group sex prior to her marriage. She reported frequently receiving religious advice from her relatives, but felt accused, disappointed, and stressed:


*“Some of my relatives care about me and my child. They often support us financially and give me advice to stay strong: ‘You have to pray a lot, God listens to people who are guilty but want to repent for their mistakes and avoid sins. This is also for the good of your son because he doesn’t deserve this [infection]’. Sometimes I take my time and think about the advice they have given me, and I feel like they are accusing me of what I did in the past. That’s why I don’t really like to visit them anymore in their house. It has been a while we don’t meet each other”.*


## 4. Discussion

Women diagnosed with HIV infection are vulnerable to a range of negative impacts, including mental health issues. This study highlights that these women experience various mental health challenges, including stress, depression, anxiety, fear, sadness, and feelings of guilt, that are in line with findings of previous studies [5,6,7,8,9,33]. As is the experience of many other WLHIV in general [8,12,34,35,36,37], a fear of death and shame due to HIV infection were two prevalent factors that contributed to mental health challenges facing these mothers.

Previous studies have found that a lack of knowledge about HIV majorly contributes to feelings of stress and fear among WLHIV following their HIV diagnosis [19,38]. This is also apparent in our findings, mothers reported not understanding HIV and feeling there was a real lack of education and resources for them, which made them feel stressed, depressed, anxious, and fearful, particularly during the early stage of their infection and when they found out that their child was infected. What our study adds to current literature is that mothers talked about not knowing whom to confide in and who to go to for information. This led to feelings of isolation and seemed to prevent them from taking necessary actions in such a challenging circumstance. In addition, there is a strong indication from our findings that these mothers were not informed or aware of any available social and health supports they could access to help themselves cope with mental health challenges and other detrimental impacts they experienced following their HIV diagnosis. This reflects how important it is to have freely available HIV-related health literacy for WLHIV and mothers of CLHIV, so they can make critical health decision [39].

Our study also reports novel findings on factors associated with mental health challenges facing mothers living with HIV, which have not been explored previously [5,6,7,8,9,33]. For these mothers, having a child diagnosed with HIV doubled the existing burden as it shifted their focus and attention from themselves to their child’s health, and made them feel even more stressed, scare, and uncertain. This enriches current evidence about double burdens people living with HIV experience, from things including: co-infection, malnutrition, and food insecurity [40,41]. Mothering CLHIV requires an enormous amount of love, devotion, and care, and takes an emotional and physical toll, especially under difficult circumstances when support is scarce. On top of this, mothers constantly worry and live with anxiety, with intrusive thoughts about themselves or their children dying, blaming themselves and feeling chronic guilt for infecting their child unintentionally [42]. Our study also identified that mothers were worried about what the future holds for their child and experienced difficulties in dealing with daily life or social activities of their CLHIV. The mothers often projected the possibility of negative consequences their CLHIV may encounter if their HIV status was made known to others. Such concerns are valid; children’s involvement in various social activities with their friends increases the possibility of their HIV status being accidently disclosed to or discovered by their friends, which could lead to negative social consequences, such as stigma and discrimination. Such concerns are underpinned by previous experiences of negative treatments towards their children and other people living with HIV by non-infected people with families, communities, and healthcare settings, which have also been reported in previous studies in different settings globally [9,15,19,20,34,35,43].

The mothers in our study also reported the difficulties and dilemma they faced when it came to dealing with questions and complaints from their CLHIV about their health condition and routine ART. Similarly, concerns about the possibility of future healthcare expenses for their CLHIV’s health needs due to poor economic and financial situations, were also supporting factors for feelings of stress and fear among these mothers. These are consistent with the previous findings [7,10,16,44], which have reported poor economic condition and healthcare expenses as determinants of depression, anxiety, fear, and worry among WLHIV. It is therefore apparent that economic or financial factors played an important role in influencing these mothers’ response to their children’s health condition, which in turn exacerbated their mental health state.

Findings from previous studies have reported the important influence of social factors such as stigma and discrimination against WLHIV, which are reflected in social rejection, social isolation, avoidance by others, on their mental health condition [5,9,13,15,19,20]. Our study adds further evidence on social factors such as unsympathetic expressions and comments from others and religious-related advice from parents and relatives, which contained negative words as contributing factors for various mental health issues experienced by the participants. It is also plausible to argue that a lack of empathy from others and negative words the mothers receive may also lead to their withdrawal from social relationships with friends and relatives, and to them missing social support from their close ones. Negative attitudes of and lack of support from families and friends have been reported as having a significant influence on access to social support and health care services by people living with HIV [34,35,43,45,46,47]. These findings demonstrate vulnerability in these participants and the role that complex socio-cultural factors play in impacting the mental health and wellbeing of women and their children living with HIV. These findings are critical and they do inform the need to address them in order to improve health and wellbeing of women and children, including families and societies that live with HIV.

The current findings have significant implications for the health sector in Yogyakarta and Indonesia as a whole. Mothers and/or housewives living with HIV represent a significant percentage of people living with HIV/AIDS in Indonesia and suffer from the impacts of HIV/AIDS. Moreover, a HIV diagnosis in mothers also negatively impacts their entire family, including their spousal relationships, social life, economic condition, and psychological state [4,38]. Such impacts may be even worse among mothers living with HIV who also have children living with HIV, thus the development of policies and interventions that address the needs of both this specific group of mothers and their children who are also living with HIV in the country is necessary and highly recommended. As the responses to HIV at both policy and practical levels in many parts of the country, including Yogyakarta, have heavily focused on halting the transmission, and less has been directed to addressing the impacts [48,49], our findings are important to inform tailored interventions that target mothers living with HIV and their children to support their needs and help them cope with various impacts they experience following their own and child’s HIV diagnosis.

### Study Limitations and Strengths

There are some limitations of the current study that need to be considered in interpreting its findings. The use of snowball sampling technique for the recruitment of the participants and the recruitment through an HIV clinic and WhatsApp group of people living with HIV may have led to the recruitment of participants from the same networks of mothers who were accessing HIV care services for them and their children. These may have led to the under-sampling of mothers who were not in the healthcare service networks who may have different stories to tell. Thus, the findings presented in this study mainly reflect experiences of a specific group of mothers. However, the strength of this study is that, to the best of our knowledge, it is an initial study that specifically involved mothers living with HIV who also have CLHIV in the context of Indonesia. Future large-scale studies involving mothers living with HIV who have CLHIV in Indonesia are warranted, if we are to gain a comprehensive understanding of their collective experiences and resulting mental health challenges, so the right supports can be put into place for the women and their children to thrive and feel supported. 

## 5. Conclusions

This study reports a range of mental health challenges, such as stress, depression, anxiety, fear, sadness, and the feelings of guilty experienced by mothers living with HIV who have CLHIV in Yogyakarta, Indonesia. These mental health challenges were associated with factors such as lack of knowledge about HIV, fear of death, shame, not knowing whom to talk with and what to do after their own HIV diagnosis, and the HIV diagnosis of their children. Other influencing factors included difficulties in dealing with their CLHIV’s social activities, and questions and complaints about HIV treatment, and concerns about the health condition of their CLHIV and negative social reactions from friends towards them and their CLHIV. Mothers in Yogyakarta do not feel supported in coming to terms with their HIV diagnosis, or that of their child, nor do they feel they have access to adequate social, financial, and medical support. Fear of stigma and discrimination is a prevalent issue for WLHIV and CLHIV, which has very negative impacts on mental health, especially for the mothers. Currently, there is very limited support for people living with HIV in general, and even more so for WLHIV and CLHIV. There is an urgent need for further research and targeted programs that support the needs of HIV-positive mothers who have CLHIV. These may include education or counselling programs for how to deal with HIV-related negative feelings, manage social interactions and activities of their CLHIV, and address their HIV-related questions and complaints. Financial support through the provision of small-scale business capital for example, would also be ideal.

## Figures and Tables

**Table 1 ijerph-19-06879-t001:** Characteristics of participants.

Mothers Living with HIV	
**Characteristics**	**N = 23**
**Age**	
21–30	10 (43%)
31–40	12 (52%)
41–50	1 (5%)
**Marital status**	
Married	15 (65%)
Widowed	5 (22%)
Never married/single	3 (13%)
**Education**	
University	3 (13%)
Senior high school	16 (70%)
Junior high school	4 (17%)
**Employment status**	
Employed	15 (65%)
Unemployed	3 (13%)
Housewives	5 (22%)
**Duration of Living with HIV (year)**	
1–5	7 (31%)
6–10	12 (52%)
11–15	4 (17%)
**Children Living with HIV**	
**Characteristics**	**N = 23**
**Age (year)**	
1–5	8 (35%)
6–10	11 (48%)
11–15	4 (17%)
**Sex**	
Male	8 (35%)
Female	15 (65%)
**Education**	
No yet	5 (22%)
Kindergarten	6 (26%)
Primary school	10 (43%)
Junior high school	2 (9%)

## Data Availability

The data presented in this study are available on request from the corresponding author. The data are not publicly available due to restrictions set by the human research ethics committee.

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
