# Peer review of "Mental Health Challenges and the Associated Factors in Women Living with HIV Who Have Children Living with HIV in Indonesia: A Qualitative Study"

_ijerph, 2022, doi:10.3390/ijerph19116879_

Round 1
Reviewer 1 Report
Thank you for the opportunity to review this manuscript. This manuscript is a qualitative study examining the factors associated with women who have children living with HIV in Indonesia. There are several points that need to be addressed.
1. I don’t see necessarily using the abbreviation of WLHIV/ PLHIV/ CLHIV, which sometimes causes confusion for the readers.
2. Line 45-46 do these unskilled workers categories include HIV-positive mothers and/ or housewives? If yes, they are not comparable.
3. Authors might consider adding more information about HIV in Yogyakarta to provide more rationale Yogyakarta is not a unique case in Indonesia, which could reduce the applicability of the study findings.
4. I have a question about the sample. As all samples are on ART, which might be expensive (method could talk a little bit about the ART like insurance coverage etc.). As such, the sample might be too restricted.
5. In addition, they are all recruited from WhatsApp, which also causes the sample might not be representative of the majority of Indonesian women.
6. The name of the interviewee should not be identifiable, please use ID or another way instead of names.
7. Discussion talks a lot about similarities/differences between the current study and previous ones. More is needed for the implication of current findings and to provide more insight into why this might be the case for CLHIV. For example, the future expenses for children’s ART treating HIV, which could the government, and society do to help them.
Reviewer 2 Report
While the introduction should situate the reader more in depth about the research problem, the rest of the paper is correct.
The sample may be small but it is valid for a qualitative study of this nature.
I also have doubts about the data analysis, more explanation of the methodology used should be taking into account.
The section on the limitations of the study would be appreciated.
Despiste of this suggestion I think that this paper has a huge potential. Thank you very much for the oportunity of rewieving this manuscript
Author Response
Reviewer 2:
- While the introduction should situate the reader more in depth about the research problem, the rest of the paper is correct.
Response:
- Despite this, there is a lack of research globally and within Indonesia about HIV-related mental health challenges and the associated factors in mothers living with HIV who have CLHIV. Whilst a few studies on WLHIV in the context of Indonesia may have overlaps with mothers living with HIV as some of the WLHIV may also be mothers (21-23), none of them has focused on mental health challenges experienced by these women. This study aims to fill this gap in knowledge by interviewing mothers living with HIV who also have CLHIV in Yogyakarta, Indonesia. Yogyakarta was chosen as the study setting due to feasibility, familiarity and the potential of undertaking the current study successfully. Exploring these factors is imperative to form an understanding of the lived experience of mothers living with HIV and their children, to better inform the development of policies and intervention programs that address the complex needs of both the mothers and the CLHIV.
- The sample may be small but it is valid for a qualitative study of this nature.
Response:
- Thank you very much.
- I also have doubts about the data analysis, more explanation of the methodology used should be taking into account.
Response:
- Data analysis steps have been made clear in the manuscript:
The comprehensive data coding and analysis were performed manually following the verbatim transcription of the interview recordings. Transcription occurred alongside the data collection process which enabled direct integration of notes taken during the interviews into each transcript. Coding and analysis were conducted in Indonesian and quotes for publication purposes were translated into English by NKF who is fluent in both Indonesia and English. To maintain the accuracy of translation, checking and rechecking the original transcripts against the translated interpretations were repeatedly conducted during data analysis process (30)]. Informed by Ritchie and Spencer’s qualitative data analysis framework, five steps were undertaken throughout data analysis process (31, 32). These included: 1) familiarisation with the data, which was performed by reading each individual transcript repeatedly. While reading the transcripts, the participants’ narratives were broken down into chunks of information, which were then labelled and commented to search for patterns and connections; 2) identification of a thematic framework which was conducted by identifying and listing key issues, concepts and ideas that emerged from the transcripts, which were then used to form a thematic framework. The identification of the thematic framework involved changing and refining the issues, concepts and ideas throughout data analysis process; 3) indexing the entire data through which data extracts in each individual transcript were indexed or coded (open coding) resulting in a long list of codes. This was followed by the identification of similar or redundant codes which were then collated (close coding) to reduce the list of codes to a manageable number. Codes that formed a logical pattern to explain a theme were grouped together under the theme or sub-theme; 4) charting the data through which all themes and their codes which had been created in the previous steps were reorganised in a summary of chart for the purpose of comparison within and across the interviews; 5) finally, the entire data were mapped and interpreted. This data analysis was non-linear and an iterative process which involved changing and refining codes and themes before we agreed upon the final themes presented in this paper. The steps undertaken in this analysis helped us manage the data in a structured and coherent manner and enhanced the rigour, transparency and validity of the analytical process (31, 32).
- The section on the limitations of the study would be appreciated.
Response:
- Thank you very much.
- Despiste of this suggestion I think that this paper has a huge potential. Thank you very much for the oportunity of rewieving this manuscript
Response:
- Thank you very much.
Reviewer 3 Report
‘The UNAIDS reports an estimated 37.7 million people living with HIV (PLHIV) worldwide in 2020’ – data need updating.
A Literature review section is missing.
What qualitative data analysis tool did you use for text mining? Any fact-checking tools? Perceptual data analysis tools? Otherwise, all you can get are subjective, uncontrolled assertions.
The manuscript requires major revisions to contextualize the merits of the study and potential uses of its methodology in future studies.
Conclusion needs to be rewritten so that only important results are brought out along with their interpretation, comparison with earlier studies, and implications in a more integrated fashion.
Round 2
Reviewer 1 Report
The authors have addressed my concern.
Reviewer 3 Report
The revised version can be published.